# Recommending Physical Activity for People with Intellectual Disabilities: The Relevance of Public Health Guidelines, Physical Activity Behaviour and Type of Contact

**DOI:** 10.3390/ijerph20085544

**Published:** 2023-04-17

**Authors:** Christoph Kreinbucher-Bekerle, Wolfgang Ruf, Astrid Bartholomeyczik, Frank Wieber, Nikolai Kiselev

**Affiliations:** 1Institute of Human Movement Science, Sport and Health, University of Graz, 8010 Graz, Austria; 2Institute of Sport Science, German University of Health and Sports, 10587 Berlin, Germany; 3Institute of Public Health, Zurich University of Applied Sciences, 8400 Winterthur, Switzerland; 4Institute of Social Psychology and Motivation, University of Konstanz, 78464 Konstanz, Germany; 5PluSport, Umbrella Organization of Swiss Disabled Sports, 8604 Volketswil, Switzerland; 6Swiss Research Institute for Public Health and Addiction, University of Zurich, 8005 Zurich, Switzerland

**Keywords:** disability, exercise, inclusion, physical activity, recommendation, sports

## Abstract

People with an intellectual disability (ID) often exhibit more sedentary behaviour and are less physically active than the general population. While previous public health guidelines on physical activity (PA) did not specifically address the needs of people with an ID, the recent updates now include this population, with recommendations similar to those for the general population. However, it is unclear whether the information about these guidelines has reached the broader public and what factors may influence their implementation. To investigate these issues, an online survey was conducted in Austria, Germany and Switzerland, which examined the (a) PA recommendation for people with an ID, (b) awareness of current guidelines, (c) participants’ own PA behaviour (IPAQ-SF) and (d) specific contact with people with an ID. Participants (*n* = 585) recommended similar levels of PA for people with an ID as for the general population, but knowledge of the guidelines did not affect their recommendation. However, participants’ own PA behaviour and context-specific contact (e.g., in family or at work) were associated with the recommended PA levels. Therefore, promoting the relevance of PA and fostering contact with people with an ID might be suitable ways to increase PA in people with an ID.

## 1. Introduction

Several studies have shown that regular physical activity (PA), sport and exercise lead to improved physical, psychological and social health or well-being for people with intellectual disabilities (IDs) [1,2,3,4]. Intellectual disability is a term used when a person has certain restrictions of cognitive functioning and skills, including communication, social and self-care skills [5]. The severity of the limitations can range from mild to severe. Common types of intellectual disabilities are Trisomy 21 (often called “Down Syndrome”), Fragile X Syndrome or Prader–Willi Syndrome, among others with non-specific causes.

The positive effects of PA on individuals with an ID was shown in several studies. A recent meta-analysis found that PA has a significant and large effect on mental health in children and adolescents with an ID and medium effects on psychological health [6]. According to the authors, the interventional settings with over 120 min per week of PA combined with therapeutic and aerobic exercises showed the strongest effect. Another meta-analysis came to the same results regarding the large positive effects of PA on the physical health of persons with an ID and the moderately large effects on psychosocial health, highlighting that strength training has strong physical benefits and that teaching movement and sport skills appears to benefit physical and psychosocial health [7]. Concurrently, it has been reported that increasing the level of PA in this population is very challenging and usually might be noticed only after time [8]. These results are supported by real-life reports [9].

Even though PA is essential for people with an ID, this population has not been considered in any national PA guidelines and public health recommendations until recently. Only in 2020 did the renewed PA guidelines from the World Health Organization (WHO) mention people with an ID for the first time [10]. As a reaction and in regard to the Convention for the Rights of People with Disabilities (UN-CRPD [11]), around 40% of countries worldwide have adapted the specific guidelines by mentioning this population [12] or are still in the process of adapting them. Among these countries are Austria, Germany and Switzerland [13,14,15]. While there are differences between these three national guidelines, such that only the Austrian national guidelines for PA explicitly mention people with an ID in every age group [13], and only the Swiss guidelines explain that adults with a disability benefit from regular PA [15], all three emphasise the importance of PA for people with an ID in general.

However, despite the profound evidence regarding the benefits of regular PA and the recent inclusion in the guidelines, people with an ID show increased levels of sedentary behaviour [16] and less PA compared to individuals without an ID [17,18]. For example, only 9% of the target population fulfilled the minimum recommendation of 150 min of moderate-to-vigorous PA (MVPA) per week [19]. Thus, the PA levels for people with an ID should be increased to approach the PA recommendations outlined in the national guidelines.

Due to the importance of the social environment in the daily life of people with an ID [20,21], relatives and professional caregivers surrounding them can act as significant counsellors for the PA behaviour of people with an ID and even influence their PA levels [22]. Empirical findings prove that the PA level of caregivers can be an indicator of their attitude towards PA and suggest that the more physically active caregivers are, the more likely they are to recommend a higher level of PA for people with an ID [23]. Conversely, if caregivers are less physically active, they may recommend a lower level of PA for people with an ID. This suggests that caregivers can act as significant counsellors for the PA level of people with an ID, and their PA behaviour can influence the level of PA they recommend for people with an ID. At the same time, the amount and quality of contact with people with an ID might contribute to the recommended PA, as highlighted by Allport’s contact hypothesis [24] and previous research showing that the degree of contact with people with an ID can influence the attitudes of this target population [25]. Accordingly, whereas stereotypes about health-enhancing PA for people with an ID might reduce the level of recommended PA [26], experiencing positive interactions and getting to know people with an ID can reduce the gap between the recommended PA for individuals with and without an ID. Nevertheless, none of these aspects have ever been investigated in German-speaking countries.

The present research aims to fill the described research gaps in the implementation of PA guidelines for people with an ID. It examines the PA level recommended for people with an ID in three German-speaking countries: Austria, Germany and Switzerland (the so-called “DACH region”). We hypothesise that participants recommend lower levels of PA than the current national guidelines and that their (a) knowledge of guidelines, (b) own PA behaviour and (c) contact experiences with people with an ID might be positively associated with the recommendations of PA for people with an ID.

## 2. Materials and Methods

### 2.1. Setting

The aim of the study was to assess how people with different forms of contact with people with an ID view them and what PA they recommend for people with an ID. The present study collected data in the three European countries of Austria, Germany and Switzerland, but as they are assumed to be similar, did not aim to compare the outcome between the countries.

### 2.2. Procedure

Data were collected using an online survey. The research team spread invitations to participate on commencement day, 24 March 2022, in Austria, Germany and Switzerland. Due to the Easter break (depending on the country and region, between the second and the last week of April), we planned to collect the data two weeks prior and two weeks after the break. Following our plan, we closed our survey on 14 May 2022.

### 2.3. Recruiting

The organizations providing support and voluntary sport or PA for people with an ID in the three countries were identified and received an email invitation with the link to the survey asking them to spread the invitation to participate among their employees, trainers, etc. The situation in the three countries is comparable based on the social systems and taking care of people with an ID, mainly in assisted facilities of social service providers. Regarding PA and sport offers, they either take place directly in the offers of the social service provider or in offers of the Special Olympics or, as in Switzerland, with nationwide external providers, such as PluSport (Volketswil, Switzerland) or Procap (Olten, Switzerland) [27]. Therefore, those organizations related to sports for individuals with disabilities placed the information about the survey and the link to it on their webpage and in their regular newsletter. Furthermore, aiming to include persons with very little or no experience and contact with people with an ID, it was written in the invitation that participants with no or minimal experience and/or contact with the population of interest were explicitly welcome to participate in the survey. The invitation to the survey was distributed via university-specific channels, as well.

Persons willing to participate (voluntary sampling) were able to activate the link, leading them to the landing page with informed consent. Only after they accepted the informed consent and confirmed their participation (opt-in) were they forwarded to the survey.

### 2.4. Online Survey

The questionnaire was developed by the authors of the present manuscript in German based on profound literature research, previous publications and the expertise of the authors in the field. The survey was validated beforehand by 11 persons with a wide variety of sociodemographic backgrounds related to sports and/or disabled sports (e.g., professionals, caregivers, trainers of people with an ID, etc.). When possible, the feedback was integrated into the final version of the survey containing 49 questions dispatched in 5 parts. Some of the questions were closed ones with yes/no answers or with 4- or 5-point Likert scales. In contrast, other questions—for example, the ones regarding a brief explanation of their PA recommendation—were open-ended. Items included for the analysis in the current manuscript are shown in the Appendix A. The five parts of the questionnaire were presented in the following order: (1) Contact with people with an ID, with context and frequency [28]; (2) Recommendation of PA for people with an ID according to public health guidelines [10]; (3) Knowledge of guidelines (WHO guidelines, UN-CRPD and current nation-specific guidelines for Austria, Germany and Switzerland), as well as related documents; (4) Own PA behaviour with the short form of the International Physical Activity Questionnaire (IPAQ-SF) [29]; and (5) Sociodemographic data, such as age, gender, work-related experience or education in the field of disability, as well as sports and exercise.

The final questionnaire was translated into English, French and Italian and backwards by professional interpreters. Additionally, each language version was cross-checked by two mother tongue employees, working in the particular language region (except English). Adjustments, if needed, were implemented.

The survey was published and completed online by means of LimeSurvey. The survey was hosted in the domain zone of the University of Graz. This software allows collecting raw data without recording personal participant information, such as the IP address, software system or region. Completing the questionnaire took around 15 min.

### 2.5. Data Analysis

To see if the sample was comparable in terms of sociodemographic data, ANOVA and Chi^2^-tests were calculated, in addition to descriptive analysis. The recommended amount of PA as a dependent variable was divided into four parts related to the intensity level: light (LPA_rec_), moderate (MPA_rec_), vigorous (VPA_rec_) and moderate-to-vigorous intensity (MVPA_rec_). These variables were not normally distributed, according to Shapiro–Wilk (LPA_rec_, MPA_rec_, VPA_rec_ and MVPA_rec_: *p* < 0.001). Therefore, non-parametric tests were used if applicable. In the first part of the analysis, recommendations were descriptively analysed and set in connection to the recommendation of muscle-strengthening activity with Spearman’s rho correlations. A MANOVA was calculated, as well, to see if there were country-specific effects on the four intensity domains of the dependent variable. The influence of the three independent variables—(a) knowledge of the guidelines, (b) own PA behaviour and (c) type of contact—was tested in a separate analysis, with a presentation of descriptive results first, then followed by inference statistics. In the knowledge part, Spearman’s rho correlations and Mann–Whitney U-tests were performed. For the participants’ own PA behaviour measured with IPAQ-SF, in reference to the IPAQ scoring protocol [30] and the analogue to the procedure used in a previous study [23], activity minutes below 10 were not counted, and the data were trimmed, with a maximum of 180 min of PA per day. Spearman’s rho correlations were used between participants’ own and recommended PA, and Wilcoxon tests were performed to see if there was a difference between participants’ recommendation and own behaviour. Finally, to test the influence of contact, a Mann–Whitney U-test was performed to see if contact had an influence, followed by MANOVA to check for any differences between the dependent variables related to the context of contact (work, sports club, family).

## 3. Results

### 3.1. Subjects

In total, *n* = 585 people fully completed the survey. Four people specified that they belonged to countries outside the scope of the investigation. Therefore, their answers, as well as those of another 64 people who completed the questionnaire only partially, were not included in the analysis. Regarding the partial completions, 9.4% of participants stopped around the recommendation of PA, 20.3% at the guidelines, 51.5% at IPAQ-SF and 18.8% at the sociodemographic data. There were no differences regarding contact with people with an ID, as well as in the recommendation of PA, but relevant data on the knowledge of guidelines, one’s own PA behaviour or sociodemographic data were missing. 

Of those final 585 people, 361 (62.0%) were female, 220 (37.8%) were male, and 1 person (0.2%) reported another gender identity. Country specifically, 318 people (54.4%) were from Switzerland, 164 (28.0%) from Austria and 103 (17.6%) from Germany. The ages ranged from 17 to 84 years (*M* = 43.33; *SD* = 14.54). Regarding educational levels, 319 (55.0%) participants reported holding a tertiary or comparable degree, 231 (39.8%) had a secondary degree (e.g., high school diploma, professional job-related education), and 30 (5.2%) had a primary school degree. The connection between their education and individuals with disabilities was confirmed by 317 people (54.2%). Furthermore, 285 (48.7%) respondents reported being in possession of a qualification (not education-related) or degree in the field of sport and exercise. 

People from the 3 participating countries did not differ in age (ANOVA: *F*_4, 572_ = 2.25; *p* = 0.062), gender (*χ*^2^ = 13.09; df = 8; *p* = 0.109), highest educational degree (*χ*^2^ = 14.63; df = 8; *p* = 0.067) or in the frequency at which their education had to do with disability (*χ*^2^ = 5.50; df = 4; *p* = 0.240). Sport-specific qualifications differed such that in Austria, 66.5% had a qualification, compared to 43.1% in Switzerland and 37.9% in Germany (*χ*^2^ = 33.47; df = 4; *p* < 0.001).

### 3.2. Recommendation of PA

#### 3.2.1. Endurance-Related and Muscle-Strengthening Recommendations 

The vast majority of 96.1% (*n* = 562) respondents would recommend PA for people with an ID. Only 3 (0.5%) would not recommend PA, and 20 (3.4%) indicated being indecisive (“don’t know” answer). The number of suggested minutes of PA for people with an ID is depicted in Table 1 for all intensities (light, moderate, vigorous and moderate-to-vigorous). Compared to these numbers, 54.6% of the respondents suggested less than the 150 min of recommended MVPA according to public health guidelines. A total of 45.4% suggested 150 or more minutes of MVPA, and 12.9% suggested 300 min or more MVPA, as is mentioned in the guidelines with additional health benefits.

Muscle-strengthening activities were suggested by 366 (62.6%) respondents. Additionally, 57 opposed this suggestion (9.7%), and 162 did not know (27.7%). Regarding those who suggested activities, the number of suggested days ranged from 1 to 7 days, with a mean of 2.57 (*SD* = 1.27) (Figure 1).

Taking together the endurance recommendation (minutes per week) and the recommendation for muscle strengthening, 177 (30.25%) recommend at least 150 min MVPA and muscle-strengthening activity. In this context, only 157 (26.84%) mentioned at least 2 days of muscle-strengthening activities. Thus, according to current public health guidelines, 73.16% did not recommend the suggested amount of physical activity for people with an ID. There is a correlation between endurance and muscle-strengthening recommendations (*r_τ_* = 0.198; *p* < 0.001; *n* = 585). The higher the probability of recommending endurance-related PA, the higher the probability of recommending muscle-strengthening activities, or vice versa. 

Multivariate ANOVA revealed that there are no country-specific differences in the endurance-related recommendation of PA for all intensity types (LPA_rec_: *F*_2,539_ = 1.29; *p* = 0.277; MPA_rec_ = *F*_2,539_ = 0.01; *p* = 0.995; VPA_rec_: *F*_2,539_ = 2.02; *p* = 0.134; MVPA_rec_: *F*_2,539_ = 0.37; *p* = 0.693). Therefore, all further results will be presented for all respondents, irrespective of their home country.

#### 3.2.2. Explanation for Recommending PA and Information Source

The open-ended question regarding the possible explanations for the suggested recommendation was answered by 67.2% (*n* = 393) of the participants. The majority of the reported answers were related to the health benefits of regular PA as a leading argument for their previous suggestion (physical, psychological and social well-being) (94; 23.9%), as the following statement illustrates: “In my opinion, sport, exercise, and physical activity usually have a positive effect on health and well-being”. Almost a fifth of the respondents for this question (19.1%; *n* = 75) connected their suggestion to the regular PA recommendations, with comments such as: “People with intellectual disabilities are just as physically capable as people without disabilities. However, they usually have a higher need for guidance and support”.

That personalization respective to age, capability or medical condition would be needed (57; 14.5%) illustrated the following quote: “I didn’t mention duration because it’s not possible to do this without knowing the person and their starting skills. As for people without disabilities, the establishment of a sports program must be thought out while considering many parameters.” Occasionally, there were other forms of feedback highlighting the importance of oneself being physically active: “Not doing sports or being physical active, I find it difficult to recommend it”.

As seen in Figure 2, the most important source for information about PA for people with an ID was the internet (77%), followed by advice from experts, such as trainers, physiotherapists, etc. (60%). In the *other* category, participants mentioned further educational training and individual experience, as well as talking to other people.

### 3.3. Knowledge of PA Guidelines

More than 50% of the respondents reported not being familiar with current guidelines related to the health-enhancing effects of PA (Table 2). Only a small number of participants mentioned being aware of the content of both national and international PA guidelines and the UN-CRPD. Furthermore, 67 of the respondents (11.5%) mentioned other health-promoting PA documents that are indirectly related to the guidelines.

There was no significant correlation between the self-reported knowledge of either the WHO guidelines or UN-CRPD in all domains of recommended PA for people with an ID (all *p* > 0.05). Country specifically, we found only tendential effects for the guidelines in Switzerland, where being more aware of the content was related to a higher recommendation in terms of minutes of vigorous (*r*_s_ = 0.110; *p* = 0.060; *n* = 292) and moderate-to-vigorous intensity (*r*_s_ = 0.107; *p* = 0.067; *n* = 292). Mann–Whitney U-tests comparing two groups (those who know with those who do not know the content of guidelines) also led to no effects, neither for the WHO guidelines nor for the UN-CRPD (all *p* > 0.05). Again, the country-specific results showed an impact in Austria for light-intensity PA (*U* = 2094.5; *p* = 0.032; *n* = 148). Thus, at least for 1 domain in 1 country, it seems that those who have subjectively higher knowledge recommend more minutes (*M* = 217.00; 95% CI = 175.52–258.48; *SD* = 160.56; Mean rank = 83.59) than those who do not know the guidelines at all (*M* = 176.14; 95% CI = 141.67–210.60; *SD* = 162.68; Mean rank = 68.30). Yet, as mentioned above, there was no general association between knowledge and the amount of recommended PA in our data.

### 3.4. Own PA Behaviour

Regarding their own PA behaviour, according to IPAQ-SF (see Table 3), 29.4% (*n* = 172) score below 150 min of moderate-to-vigorous PA (MVPA), which means that 70.6% fulfil endurance-related public health guidelines. Overall, 45.6% (*n* = 267) also reported 300 min or more of MVPA. Note that these results are only related to the minutes of PA. Participants were not asked if they performed muscle-strengthening activities, as recommended in the guidelines. Thus, the reported amounts of activity most likely underestimated the amount of general activity.

Related to the own PA behaviour of all respondents, there is a correlation with the recommended PA for people with an ID (MPA: *r*_s_ = 0.174; *p* < 0.001; VPA: *r*_s_ = 0.187; *p* < 0.001; MVPA: *r_s_* = 0.209; *p* < 0.001). Thus, a higher own PA behaviour is related to a higher recommendation of PA for people with an ID, or vice versa. Furthermore, we compared the own PA behaviour with the recommended PA minutes with Wilcoxon Z-tests, which revealed that the own PA behaviour is significantly higher than the recommendation for people with an ID for all intensity domains (MPA: Mean rank *=* 277.97 vs. 190.70; *Z* = 9.101, *p* < 0.001; VPA: Mean rank: 272.00 vs. 158.92; *Z* = 12.279, *p* < 0.001; MVPA: Mean rank = 296.90 vs. 171.41; *Z* = 122.992, *p* < 0.001).

### 3.5. Context-Specific Contact with People with an ID

Overall, 434 (74.2%) of the respondents mentioned that they have direct contact with people with an ID. Another 75 (12.8%) had contact in the past, whereas 76 (13.0%) never had contact with people with an ID. Those who currently have contact did not differ in their general recommendation for PA in all four intensity domains compared to respondents having no contact with people with an ID (LPA: *U* = 23,997; *p* = 0.060; MPA: *U* = 25,051.5; *p* = 0.223; VPA: *U* = 23,972; *p* = 0.055; MVPA: *U* = 24,532; *p* = 0.125). The contact mainly occurred in the working context, in which half of the participants had regular contact with people with an ID, followed by contact in a sport club or family-related setting (see Table 4). The majority of respondents rated their contact with people with an ID as close (251; 49.3%) or very close (91; 17.9%).

In a further step, PA levels were analysed for the three most-mentioned contexts of contact (work, sport club and family). Therefore, the answers for “sometimes” and “frequently” were taken together and contrasted with “never” with a multivariate ANOVA and the four domains of PA (light, moderate, vigorous and moderate-to-vigorous) as dependent variables. There was a significant main effect for the family context in the recommendation of moderate (*F*_1,386_ = 5.705; *p* = 0.017; *n*^2^ = 0.015), vigorous (*F*_1,386_ = 4.876; *p* = 0.028; *n*^2^ = 0.012) and moderate-to-vigorous intensity (*F*_1,386_ = 6.497; *p* = 0.011; *n*^2^ = 0.017), but no significant main effects for the context of work or sport club (*p* > 0.05). Those who have regular family-related contact recommended higher values of moderate (*M* = 131.52, *SD* = 110.55) and moderate-to-vigorous intensity (*M* = 195.33, *SD* = 165.58) compared to those with no contact (MPA_rec_: *M* = 106.68, *SD* = 85.28; MVPA_rec_: *M* = 159.72, *SD* = 134.76). An interaction effect was found for family and work in vigorous (*F*_1,386_ = 7.552; *p* = 0.006; *n*^2^ = 0.019) and moderate-to-vigorous intensity PA (*F*_1,386_ = 4.317; *p* = 0.038; *n*^2^ = 0.011). All other interaction effects were not significant (*p* > 0.05). The data suggest that high contact in family and work leads to a lower recommendation of PA than having high contact within the family and no work-related contact (Figure 3).

## 4. Discussion

The current paper investigated the amount of physical activity (PA) recommended for people with intellectual disabilities (IDs) by individuals with different forms of contact with people with an ID. Thereby, it addresses the research and knowledge gap on the implementation of PA recommendations for people with an ID. Specifically, it aims to explicate whether the PA recommendations for people with an ID that are now included in current PA guidelines are reflected in the recommendations of PA regarding light, moderate, vigorous and moderate-to-vigorous intensity. We hypothesised that three variables might influence these recommendations, namely, the knowledge of guideline recommendations, participants’ own PA behaviour and context-specific contact with people with an ID. As there were no country-specific differences in all outcome variables, except some minor effects related to the knowledge of PA guidelines, all results are reported overall.

### 4.1. PA Recommendations for People with an ID in General

The vast majority of the participants in the current investigation recommended PA for people with an ID, but still, more than half of all respondents recommended less than the amount of 150 min of moderate-to-vigorous intensity per week that is listed in public PA guidelines [10]. Moreover, there is a higher uncertainty in the recommendation of muscle-strengthening activities. Therefore, even fewer participants (62.6%) recommended sufficient muscle-strengthening training, and only 26.8% correctly reported both public health recommendations (150 min MVPA and at least 2 days of muscle-strengthening activities). This is partly comparable to other investigations of the general population [31,32,33], in which a very low amount of people correctly reported the guidelines, and high subjective norms towards PA were shown [34].

As an explanation for the recommendation, most respondents mentioned health benefits of regular PA [1,2,3,4], pointing to a high awareness of the benefits of regular PA. Furthermore, a fifth of the respondents linked their recommendation for people with an ID to a recommendation for the general population or the fact that one should not differ between people with and without an ID. Although the same amount should be recommended, according to 20% of the respondents, it is mentioned that people with an ID need more support, such as transportation, guidance, etc. [35]. Accordingly, respondents mention, as well, that it is hard to recommend PA in general, and any personalization in the recommendation depending on age, capability or medical condition is necessary, which is addressed in current PA recommendations [10]. As a source for their recommendation on PA for people with an ID, information is mainly derived via the internet, followed by experts, such as trainers or physiotherapists, and articles or journals.

### 4.2. Knowledge of PA Guidelines

Regarding the knowledge of public health recommendations for PA, 56.9% of the people surveyed stated that they were unfamiliar with the current WHO PA recommendations [10]. It seems that the level of knowledge has no influence on the recommendation. This becomes even more evident in the fact that participants from the three countries did not differ in their recommendations, although current guidelines in the three countries differed in mentioning people with an ID [13,14,15]. Empirical findings suggest that the knowledge of the guidelines is also related to one’s own PA behaviour [33]; those who are fulfilling the guidelines themselves have a better knowledge of them [31] or eventually have a better reference to their own PA behaviour. Although we asked for PA recommendations in minutes per week for all intensity levels, in-depth knowledge is missing as to how their answers are composed. Suppose a recommended value for PA is related to the reference in the guidelines or comparable documents. In that case, one’s own PA behaviour or estimations via simple heuristics must be deepened in further investigations. A first indication was given with an open-ended question about the explanation for the recommendation. Whereas the benefits of regular PA were mentioned by most of the individuals in the current investigation, around 20% of the participants clearly stated that the recommendation should be the same irrespective of disability, which supports the idea of the recent update of public health-related guidelines on PA [10,13].

Still, it should be discussed how the knowledge of these guidelines can be increased and transferred to the general population [36] and to people with an ID themselves. Furthermore, it should be discussed why PA guidelines seem to have a low impact on the actual PA recommendations of individuals working with people with an ID and the general public, as is indicated in the current investigation. PA recommendations are necessary as guidance for researchers, practitioners and politicians. They can guide as a reference and help to draw conclusions and practical implications or interventions. It is more important to incorporate PA and sport offers in the daily life of people with an ID and also to be able to initiate change in other people, highlighting the relevance of health-enhancing PA [37].

Next to the knowledge of PA guidelines in general and specifically to a country, we asked for disability-specific knowledge in reference to the UN-CRPD [11]. Although the relevance of PA is only indicated in this document, as people with disabilities should be given equal rights in participating in sport and leisure activities, people who do not regularly work with people with an ID showed higher recommendations of PA related to a higher knowledge of the UN-CRPD. This suggests the need for other supportive documents highlighting the relevance of PA for people with an ID in relation to general regulations or guidelines for the target group next to the promotion of the beneficial effects of regular PA.

### 4.3. Participants’ Own PA Behaviour

As it was found in a similar investigation with caregivers of people with an ID [23], it seems to be that one’s own PA behaviour might be important for the recommendation of PA. People who are more physically active in all intensity domains recommend more PA for people with an ID or vice versa. Thus, one’s own PA behaviour is a crucial predictor for a recommendation. However, one’s own PA behaviour is still higher than what the participants of the current and former investigation [23] recommend to others. One reason could be that there is low confidence in recommending a sufficient amount of PA to people with an ID. Another reason could be the overestimation of one’s own PA behaviour compared to the recommendation [38] or the fact that people find it hard to differentiate between different intensities of PA [39]. Empirical investigations focussing on qualitative data of how the recommendation for PA for different target groups is built and which motives are behind it could help to shed more light on this connection. The indications of the current analysis, in which participants reported that they find it hard to recommend PA if they are not active themselves, point in a possible direction. In this context, PA offers not only for people with an ID, but also for their caregivers, could be a promising method [40].

### 4.4. Context-Specific Contact with People with an ID

Furthermore, contact with people with an ID, in general, did not seem to have an influence on the PA recommendation, but it seems to be a relevant factor if it is context specific, for example, within the family or work related. These findings seem to be in line with the contact hypothesis of Allport [24] and disability-specific findings [25], at least for contact within family settings. People who have regular family-related contact with people with an ID recommend more minutes of PA in moderate, vigorous and moderate-to-vigorous intensity compared to people with no family-related contact. The role of the family in connection to PA is frequently considered in children [41] and adults [42], but mainly as a barrier to increasing the PA levels of people with an ID or participating in specific offers [43]. Here, we found a positive influence of contact within the family for the recommendation of PA to people with an ID.

Surprisingly there is no difference in the general recommendation of PA in the groups with regular compared to irregular work-related contact and for contact in sport clubs. Moreover, our data suggest that work-related contact alone is insufficient for a proper recommendation of PA. Nevertheless, in the work-related context, it seems to be the case that more contact leads to a decrease in suggested minutes, especially with vigorous-intensity PA. One reason could be that in a work-related context, participants referred to people with severe forms of IDs compared to other contexts [44]. Empirical findings suggest that caregivers of people with an ID hardly ever recommended them to be physically active at vigorous intensities due to the potential health-related concerns [23], although it is feasible for people with an ID and also with cardiovascular-related risk factors [45].

When considering the interaction of contacts between family and work, it should be noted that an exchange of family members and caregivers would be a proper approach to increase the probability of people with an ID being physically active [46]. Therefore, it is mandatory to create awareness of the importance of health-enhancing PA, not only in families and relatives [41], but also in people working as social service providers [23]. Only through higher awareness and a closer collaboration of all relevant actors (including people with an ID themselves) can turning existing barriers into facilitators for regular physical active and avoidance of sedentary behaviour in the target group [47]. This could lead to a higher awareness of individuals working with people with an ID related to the positive effects of PA. Thus, caregivers should be supported in theory-driven approaches connected to relatives and experts [48]. Theory-based interventions are recommended because they are effective in PA promotion, irrespective of the personal and medical backgrounds [49,50,51].

## 5. Limitations

With respect to our sample, the lack of a representative sample limits the external validity of the results. Whereas the population of individuals who work with people with an ID was approached systematically via the relevant professional organisations and stakeholders, participants from the general public were a convenient sample.

In line with a potential self-selection bias, participants in the current investigation had intense contact with people with an ID and were highly physically active, according to self-report. This seems to be a general limitation when conducting investigations linked to specific topics [52]. Only 25.8% of participants mentioned they have no contact. In total, 70.6% fulfil the criteria of PA recommendations according to self-reports, which is higher than in the general population (less than 50% in the normal population [53]). However, a self-reported PA questionnaire (IPAQ-SF) tends to overestimate MVPA by an average of 84% [38]. The convenient sample and unequal sizes of the samples from the three participating countries limit the external validity and informative value of the cross-country investigation.

Furthermore, we only asked for recommendations for people with an ID, not for the general population. This does not allow us to compare if participants would make a different suggestion for others compared to people with an ID. Still, the recommendation values are similar to the findings for the general population or even higher [54]. Moreover, the fact that 20% of the respondents in our data mentioned, without being asked, that they would recommend the same amount of PA minutes for people with an ID as for the general population can be seen as the first indicator. Therefore, we need more empirical investigations in these directions, especially in terms of implementing into practice the public health guidelines of PA for specific target groups.

We did not ask for the muscle-strengthening activities of participants, which prevents the examination of the influence of participants’ own muscle-strengthening activities on their muscle-strengthening activity recommendations for people with an ID. However, according to our data, as the recommendation of PA and muscle-strengthening activity shows a similar positive correlation to their own, PA behaviour can be expected as follows: individuals who engage in more muscle-strengthening activities would recommend higher levels of muscle-strengthening activities to people with an ID than individuals who do not engage in muscle-strengthening activities [55].

Finally, the self-reported questionnaire was explicitly given to people without an ID, asking for their recommendations for people with an ID. People with an ID and their views were not included in the current investigation. Therefore, future studies on PA should involve the perspectives of people with an ID in adapting and preparing materials and methods for the target group (e.g., easy-to-read language, visualisation).

## 6. Conclusions

Regular PA has manifold beneficial effects on all domains of health and well-being, irrespective of disability. However, people with an ID are rather inactive and do not attain the recommended levels of health-enhancing physical activity (HEPA). The present study points to concrete ways in which PA for people with an ID can be improved. Particularly promising is to get in touch with people with an ID through PA, either through their social environment or professional offers. Still, more empirical data in this field are required to see which amount of PA is necessary for people with an ID and which other factors influence their PA behaviour. Therefore, longitudinal examinations of whether and how professionals receive the guidelines, the public and people with an ID are highly recommended. Most importantly, people with an ID must be involved in a co-production process to increase their PA knowledge and behaviour.

## Figures and Tables

**Figure 1 ijerph-20-05544-f001:**
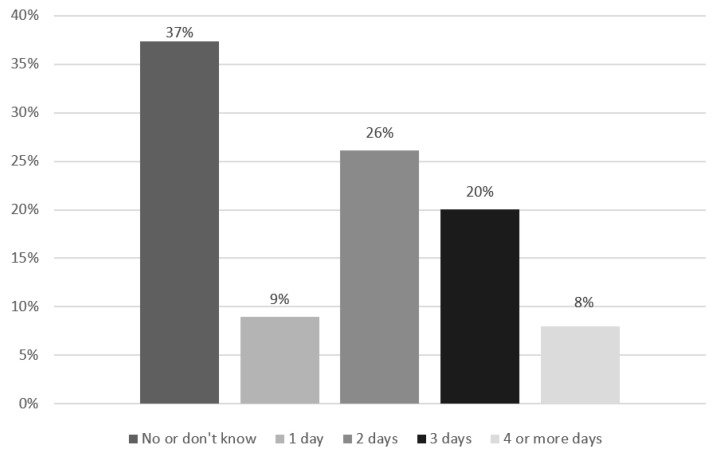
Recommended days of muscle-strengthening activities per week.

**Figure 2 ijerph-20-05544-f002:**
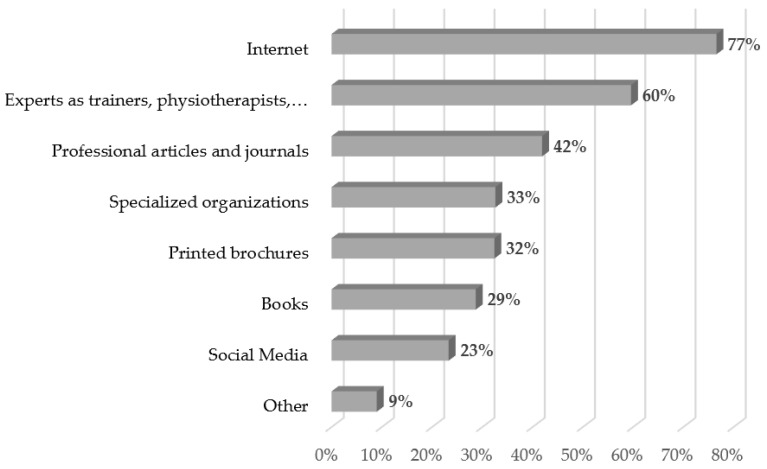
Sources for information about PA for people with an ID in % (multiple answers possible).

**Figure 3 ijerph-20-05544-f003:**
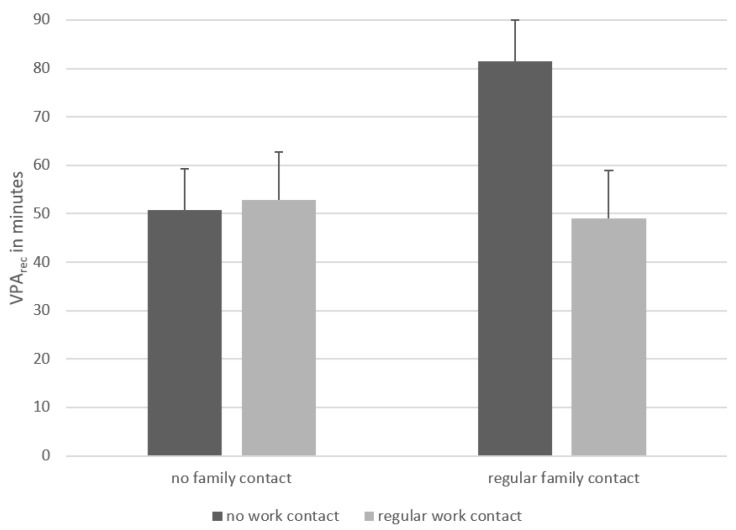
Interaction effect of contact within family and related to work in vigorous intensity.

**Table 1 ijerph-20-05544-t001:** Mean, Median and SD of suggested minutes of PA per intensity.

Intensity of PA	Range	Mean	SD	Median
LPA_rec_	0–960	208.36	176.66	150
MPA_rec_	0–600	110.42	88.04	90
VPA_rec_	0–630	54.60	62.81	30
MVPA_rec_	0–1050	165.01	137.14	120

Abbreviations: PA, physical activity; LPA_rec_, recommendation of light-intensity PA; MPA_rec_, recommendation of moderate-intensity PA; VPA_rec_, recommendation of vigorous-intensity PA; MVPA_rec_, recommendation of moderate-to-vigorous-intensity PA.

**Table 2 ijerph-20-05544-t002:** Frequency of self-reported knowledge of the specific guidelines.

Guidelines	Do Not Know (*n*; %)	Know Parts of the Content (*n*; %)	Know the Content Approximately (*n*; %)	Know the Most Important Parts of the Content (*n*; %)	Know All the Content (*n*; %)
WHO, 2020 (*n* = 585)	333; 56.9%	88; 15.0%	81; 13.8%	72; 12.3%	11; 1.9%
UN-CRPD, 2006 (*n* = 585)	291; 49.7%	85; 14.5%	85; 14.5%	95; 16.2%	29; 5.0%
Austrian, 2020 (*n*_a_ = 164)	97; 59.1%	27; 16.5%	20; 12.2%	15; 9.1%	5; 3.0%
German, 2016 (*n*_a_ = 103)	73; 70.9%	12; 11.7%	8; 7.8%	9; 8.7%	1; 1.0%
Swiss, 2017 (*n*_a_ = 318)	196; 61.6%	43; 13.5%	41; 12.9%	34; 10.7%	4; 1.3%

Abbreviations: *n*_a_, participants in the respective countries.

**Table 3 ijerph-20-05544-t003:** Mean, Median and SD of own PA behaviour of the respondents.

Intensity of PA	Range	Mean	SD	Median
MPA_own_	0–1260	224.15	271.15	120
VPA_own_	0–1260	144.99	171.53	90
MVPA_own_	0–2520	369.14	359.19	260
Walking_own_	0–6300	501.18	766.39	210
Sitting_own_	0–999	316.46	188.21	300

Notes: All categories are related per week, except sitting per day. Abbreviations: PA, physical activity; MPA_own_, own moderate-intensity PA; VPA_own_, own vigorous-intensity PA; MVPA_own_, own moderate-to-vigorous-intensity PA.

**Table 4 ijerph-20-05544-t004:** Frequency of self-reported contact with people with an ID in different context situations.

Context of Contact	Never (*n*, %)	Rarely (*n*, %)	Sometimes (*n*, %)	Frequently (*n*, %)
Work	170 (29.1%)	54 (9.2%)	55 (9.4%)	306 (52.3%)
Sport club	410 (70.1%)	52 (8.9%)	77 (13.2%)	46 (7.9%)
Family	426 (72.8%)	74 (12.6%)	31 (5.3%)	54 (9.2%)
Neighbourhood	425 (72.8%)	102 (17.4%)	46 (7.9%)	12 (2.1%)
Friends	439 (75.0%)	90 (15.4%)	35 (6.0%)	21 (3.6%)
Leisure time	444 (75.9%)	103 (17.6%)	30 (5.1%)	8 (1.4%)
Non-sport club	487 (83.2%)	58 (9.9%)	23 (3.9%)	17 (2.9%)
Education	514 (87.9%)	32 (5.5%)	17 (2.9%)	22 (3.8%)
Other	512 (87.5%)	44 (7.5%)	15 (2.6%)	14 (2.4%)

Notes: “Rarely” was additionally operationalised with once a month, “sometimes” with once a week and “frequently” with several times a week (see also the whole questionnaire in Appendix A).

## Data Availability

The data and materials associated with the current study are available from the corresponding author on reasonable request.

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
