# Peer review of "Recommending Physical Activity for People with Intellectual Disabilities: The Relevance of Public Health Guidelines, Physical Activity Behaviour and Type of Contact"

_ijerph, 2023, doi:10.3390/ijerph20085544_

Round 1

Reviewer 1 Report

The manuscript is interesting however, slightly confusing to read. Please refer to the comments attached. 

Thank you 

Author Response

We want to thank the reviewer for the many helpful and thoughtful suggestions on how to improve the manuscript. Based on those, we revised the introduction, specified both methods and analysis, and clarified some points the discussion. Our adaptions (via track changes) will be explained according to the comments. We gladly addressed them as follows:

"Rewrite Abstract and check Objectives"

We tightened the abstract as good as possible and checked back with the objectives in the end of the introduction and in subchapter 2.1

“No need as an introduction as the title is on people for ID. Directly introduce the research based on the selected population.”
Now, we avoid mentioning the positive aspects of PA in general and directly startet with the selected population of people with ID.

 “Need to explain all type of ID and not only mentioned one.”
We added other types of ID as well and mentioned that it can contain unspecific causes.

 “Please re-write to improve the fluency and to add citation [paragraph starting line 58].”
We agree that this paragraph was confusing and lacks citation, which was now rewritten and backed up with citation.

 2.1.: The objective here does not concurrent with the ones mentioned in the abstract. Please revise
We revised both abstract and this specific paragraph setting.

 "Does the study receive any ethical clearance from all the 3 countries by the related professional bodies?

Thank you for pointing out this important issue. The institutional review board approval is placed in the statements section right after the conclusion. It is supposed to comprise all three countries.

"How was the sampling method done?"

The sampling method used in this study corresponds to voluntary sampling. We have now added this description, as well as additional information, to subchapter 2.3, "Recruiting".

“Who are they [people asked for pre-testing]? are they professionals?”

We now explained in more details whom we asked for pre-testing.

 "Based here, it seems that some of the questions where adapted from previous published studies. Thus, does the final questionnaire been pre-test to be used in the targeted sample?"

We appreciate your question. We have now clarified this aspect more explicitly at the beginning of subchapter 2.4, "Online Survey".

"In my opinion, since this is a cross sectional study using questionnaire, a sample size calculation is needed to be able to proof the Power of the sample "

Indeed, it is common to request a sample size calculation for a cross-sectional study. However, in certain cases, a sample size calculation may not be appropriate. For example, when a study has an exploratory nature or when the sampling method is based on participant availability and willingness to participate. In our study, both of these factors were present. Therefore, a sample size calculation would not be appropriate for this paper.

I think this 4 people need to be excluded as it is not concurrent with the Objective.

We agree and excluded those 4 people from other countries. All analysis were performed again without those 4 people, leading to slightly different numbers but similar pattern and results.

Present mean and standard deviation in the tables

Thanks for that note, we added SD instead of 95%CI in the tables 1 and 3. 

Personalization instead of Individualization?

Yes, we agree on that and changed it accordingly.

 "3.5.: Do you think that people who never had any contact or experience working with PwID will give irrelevant outcome to the questionnaire comparing to the people who is working and have experience working with PwID?

We appreciate the question. However, we are unsure if we understand this question's context. In the results (subchapter 3.5.), we compare the answers of the persons with no contact with PwID to the responses of other groups of participants. Moreover, persons with no contact with PwID can serve as indicator of the knowledge and attitudes of the general population. Given that, we do not think that their answers are irrelevant.

Where there differences of outcome between the 3 countries? The discussion is not directly answering the objectives

Initially it was not our aim to compare those 3 countries. Instead we wanted to have a broad overview in this specific region of “German”-speaking alpine countries with strong cross-cultural similarities among the region. We did mention this already in the results but highlighted it as well in the discussion section. Please see the discussion as well below.

"In my opinion, the conclusion is incoherent to the objectives. Since the study is comparing between 3 European countries, the discussion and conclusion lacked to emphasize on the comparison as these 3 countries may behave differently towards the subject matter based on their social, cultural and behavioral background. "

Comparing the three countries is an intriguing idea. However, the topic of inclusion and the stage of its progress in society varies and can be a politically sensitive issue. Consequently, any comparison between countries related to inclusion may be overhyped by stakeholders in those countries. As researchers, we aim to avoid political and controversial discussions. Moreover, the rather small samples in the respective countries prevents a comprehensive comparison and drawing valid conclusions. Thus, we did not compare the countries. However, we agree that it might be of value if later studies do this.

Reviewer 2 Report

This is a very well researched and well written article, highlighting the importance of physical activity for persons with intellectual disability, an aspect of life that is often neglected. 

The introduction provides an excellent summing up of the background to the research. The methodology section clearly explains the rationale for the research and the methods used to conduct it and to analyse the date.

The results section presents a comprehensive and rigorous analysis of the data gleaned from the surveys. This analysis is reinforced by the discussion of the results. 

The recommendations made are appropriate and should be of benefit for persons with intellectual disability.

There are only two comments:

- avoid using 'PwID' to refer to persons with intellectual disability. It is important to avoid using initials and acronyms to refer to people. Instead, the term 'persons with ID' can be used. 

- an inherent limitation of the research is the lack of voice of the persons with intellectual disability themselves. It is appreciated that tapping into this voice was beyond the scope of this particular study. However, the wealth of disability studies research shows the fundamental importance of understanding the experiences of persons with disability from their own perspective. Therefore, it is recommended that this lack of voice is included in the Limitations section.

The authors may also wish to consider following up this interesting research with a study conducted with persons with intellectual disability involved in physical activity. 

Author Response

We want to thank the reviewer for the many helpful and thoughtful suggestions on how to improve the manuscript. Based on those, we changed the term PwID to persons with ID and enlarged our limitations. Our adaptions (via track changes) will be explained according to the comments. We gladly addressed them as follows:

"- avoid using 'PwID' to refer to persons with intellectual disability. It is important to avoid using initials and acronyms to refer to people. Instead, the term 'persons with ID' can be used."

Thank you for your advice. We have now adjusted the naming throughout the manuscript.

"an inherent limitation of the research is the lack of voice of the persons with intellectual disability themselves. It is appreciated that tapping into this voice was beyond the scope of this particular study. However, the wealth of disability studies research shows the fundamental importance of understanding the experiences of persons with disability from their own perspective. Therefore, it is recommended that this lack of voice is included in the Limitations section."

We appreciate that you brought up this aspect. We did not initially emphasize the importance of including participants with intellectual disabilities in our study. We have now included this issue in the limitations section of the manuscript.

"The authors may also wish to consider following up this interesting research with a study conducted with persons with intellectual disability involved in physical activity."

Thank you for this idea, which is totally in the direction our research is focussing at the moment. Currently, we have several participatory research projects that aim to investigate the perspectives of people with disabilities and hope to present some results in the near future.

Reviewer 3 Report

The study looks good overall. The manuscript is titled, "Recommending Physical Activity for People with Intellectual Disabilities: The Relevance of Public Health Guidelines, Physical Activity Behavior and Type of Contact". The study lacks a representative sample and thus the validity of the same could be relooked at. Would be ideal if the authors considered studying the impact of the survey on at least a few participants with a PwID to validate it. Also, self-reporting is a minor limiting factor.

Author Response

We want to thank the reviewer for the many helpful and thoughtful suggestions on how to improve the manuscript. Based on those, we made some adjustments and enlarged our limitations. Our adaptions (via track changes) will be explained according to the comments. We gladly addressed them as follows:

 “The study lacks a representative sample and thus the validity of the same could be relooked at. Would be ideal if the authors considered studying the impact of the survey on at least a few participants with a PwID to validate it.”

We thank for this important comment and we are currently expanding our idea to people with ID. The instrument as it is can’t include people with ID because the language and procedure might not understandable for them (without assistance). We are in a process of adapting for example the IPAQ in the direction of comprehensibility for people with ID, but to include this in the current paper would far exceed the paper and its original aim.

Also, self-reporting is a minor limiting factor."

We agree that this is a limiting factor as well and already mentioned the problems related to self-reported physical activity behaviour, which is mostly overestimated by people. Furthermore we added the minor limitation with self-reporting in general as well in the limitation section.

Reviewer 4 Report

The authors have administered a significant study that concerns cultivating intellectually disabled people with physical activity. Although the survey has considerable merits, potential comments must be considered to enhance the manuscript's quality.

Abstract: The authors must mention the number and identity of the study participants who attempted to complete the online questionnaire.

Introduction: In this section, the authors haven't precisely shown any literature review under a different section that discusses the application of physical activity on mentally challenged people. Thus, I kindly request the authors to divide the introduction section.

Method: The author must state their research design to conduct the online survey. What sampling technique? How were the participants approached? Who exactly are the participants? What kind of consent (written/ Oral) was obtained?

Since the authors designed the instrument, it is more required to address the reliability of the instrument.

Author Response

We want to thank the reviewer for the many helpful and thoughtful suggestions on how to improve the manuscript. Based on those, we made some adjustments on abstract, introduction and methods. Our adaptions (via track changes) will be explained according to the comments. We gladly addressed them as follows:

"Abstract: The authors must mention the number and identity of the study participants who attempted to complete the online questionnaire."

We brought up this point in a footnote, in which we mentioned how many participants droped out and at which stage in the questionnaire. We are convinced that this is a good solution not disturbing the flow of reading but still explaining it.

"Introduction: In this section, the authors haven't precisely shown any literature review under a different section that discusses the application of physical activity on mentally challenged people. Thus, I kindly request the authors to divide the introduction section."

Thank you for this suggestion. We have added a section on the effects of physical activity on individuals with intellectual disabilities. We recognize that this is an important issue that should be emphasized in the manuscript.

 “Method: The author must state their research design to conduct the online survey. What sampling technique? How were the participants approached? Who exactly are the participants? What kind of consent (written/ Oral) was obtained?"

Thank you for your inquiry. Our sampling method utilized voluntary sampling, which is discussed in the extended subchapter "2.3. Recruiting". The participant description is presented in the results section. Since we cannot anticipate the sample description prior to conducting the survey, we are hesitant to include it in the methods section. Instead, we consider it a result and have placed it in the results section.              

Since the authors designed the instrument, it is more required to address the reliability of the instrument.

We agree that the reliability of an instrument, specifically internal consistency (retest reliability cannot be reported here), is critical to its quality. However, the designed instrument is mainly based on reliable and valid instruments as the IPAQ or questions related to the contact to people with ID (at least the German version). We only added questions regarding the recommendation of PA (also based on the literature of WHO recommendations) and about the knowledge of these guidelines. Therefore we think a reliability analysis is not compulsary in this context. 

Round 2

Reviewer 1 Report

Unfortunately, the authors have not adequately addressed the points raised in the previous round of reviews. Please review the previous comments and ensure that a careful response is provided to each comment.

Author Response

We want to thank the reviewer again for the conscientious remarks made according to our manuscript. We agree with the reviewer that we did not address all raised points properly. Therefore, we thoroughly checked all comments of the reviewer and revised our answer to the reviewer (see attached document with track changes according to the answers in the first round). Furthermore, we made some adjustments in the manuscript, especially in the abstract and paragraph starting in line 81. We think the manuscript now increased in quality and we are looking forward to a final decision. 

Round 3

Reviewer 1 Report

Thank you for the further clarification.